# Older adults' suggestions of research topics on ageing well in urban environments – A participatory study

**Philip Oeser** [1]*, **Nora Bruckmann**[1], **Paul Gellert**[2], **Wolfram J. Herrmann**[1]

1 Institute of General Practice and Family Medicine, Charité - Universitätsmedizin Berlin, Berlin, Germany,
2 Institute of Medical Sociology and Rehabilitation Science, Charité – Universitätsmedizin Berlin, Berlin, Germany

* philip.oeser@charite.de

**Data Availability Statement:** The data on Berlin neighborhood statistics is publicly available. Amt für Statistik Berlin-Brandenburg, Statistischer Bericht A | 16 – hj 2 / 20, available online at: [XLSX]

## Abstract

### Background

Ageing societies and urbanization are global phenomena that pose new challenges for care delivery. It is important to create a scientific evidence base to prepare for these changes. Hence, the aim of our study was to assess which research agenda older adults living in an urban environment in Germany suggest.

### Methods

A total of 1000 participants aged 65 years or older from five different neighborhoods of Berlin were randomly chosen and were sent a single item questionnaire allowing them to freely propose research topics regarding ageing well in the city. Codes were developed inductively and clustered into categories. In a second stage, these results were discussed with the participants and local stakeholders in a workshop and video calls.

### Results

102 persons suggested 18 research topics in 6 categories: health, living environment, social issues, mobility, and accessibility to information and communication. Proposed research topics ranged from accessibility of health care, green spaces and recreational means to social involvement and loneliness.

### Conclusion

There is a substantial interest of older adults for research regarding their living situation. Research projects and local urban planning committees are encouraged to invite older adults to participate and integrate their perspectives suggested by older adults.

**Funding:** The project on which the article is based was supported by the Berlin Institute of Health (BIH) QUEST Center for Responsible Research. It was funded with 17,810 € in total over a duration of six months by the QUEST Grant for Patient and Stakeholder Engagement. The funders had no role in study design, data collection and analysis, decision to publish, or preparation of the manuscript.

**Competing interests:** The authors have declared that no competing interests exist.

## Introduction

Ageing societies and urbanization are two global developments which pose challenges for healthy ageing in urban areas. The share of persons 70 years or older will double globally from 5.9% in 2020 to 11.3% in 2050, a shift that is also pronounced in Germany with an estimated increase from 15.9% in 2020 to 23.6% in 2050 [1]. An ageing population changes the demands for healthcare, with the focus shifting from acute care to long-term care, chronic conditions, and non-communicable diseases. Furthermore, the worldwide urban population is expected to increase from 4.22 billion in 2018 up to 6.68 billion in 2050, while the worldwide rural population is declining from 3.41 billion people in 2018 to 3.09 billion people in 2050 [2]. Germany already has a high degree of urbanization and ranks 14th in the decline of rural population until 2050. An urban living environment is associated with a high population density and more diverse population regarding gender, migration background and socio-economic status (SES) [3]. Healthcare and social care in urban areas are at the same time highly fragmented with several different providers available. Making cities age-friendly is becoming an important research area with a high public relevance [4].

What do older adults, the people who are directly affected, have to say about ageing well in urban environments? In a qualitative study from the United States based on interviews with community-dwelling adults older than 60 years, self-acceptance, self-growth and the pursuit of active engagement were defined as fundamental to ageing successfully [5]. A qualitative study with older adults on facilitators and barriers to growing old at home named physical and mental health, family environment and financial stability as key elements for quality of life while criticizing not being taken into account as a demographic by society, especially when living in an urban environment [6]. Reciprocal and trusting relationships with neighbors were the basis for satisfaction in another study on solitary older women's perspectives on their residential living area and its impact on health and wellbeing [7]. Based on data from the Belgian Ageing Studies, it was shown how the physical environment can positively influence feelings of safety in older adults when the neighborhood is adapted to their physical needs [8]. A recent qualitative study in Germany on dementia risk reduction in urban environments, involving older adults and stakeholders, explored different perspectives on designing urban environments that support older adults' needs to promote brain health, and encourages public policy to involve community members as co-creators for these spaces [9]. In an Australian study by the National Ageing Research Institute, the question of what older people want from healthcare was extensively assessed using a mixed-methods approach with focus group interviews and an online survey, emphasizing on topics surrounding health care [10].

Common to all these studies is that older adults merely participated in the research process and did not shape the research agenda themselves. Research on projects that allow older adults, as laypersons, to specifically define research topics is limited. Thus, the aim of our study was to assess which research agenda older adults living in an urban environment in Germany suggest.

## Methods

This project had a participatory design following two stages: a survey stage and a workshop stage. First, we selected five different neighborhoods in Berlin to conduct the project. For the selection process, we used publicly available data from the Senate Department for Urban Development and Housing Berlin (Senatsverwaltung für Stadtentwicklung und Wohnen Berlin), and the Office of Statistics Berlin-Brandenburg (Amt für Statistik) [11,12] with an extensive insight in their demographics (number of inhabitants, share of inhabitants 65 years and older, SES, migration background). The selection was conducted jointly with local authorities, agreeing on neighborhoods with a heterogeneous population of older adults regarding

socioeconomic status and migration background. Key figures of the five neighborhoods are presented in Table 1.

For the survey phase, we inquired at the Agency for Civil and Regulatory Affairs in Berlin (Landesamt für Bürger- und Ordnungsangelegenheiten, LABO) to randomly choose 200 citizens in each of these neighborhoods. Inclusion criteria were age of 65 years or older (the age of retirement in Germany), no active legal guardianship, and registration of main residency in the respective neighborhood. The study information and a single open-ended item was developed and pre-tested with other researchers and senior representatives from different neighborhoods in Berlin. The single open-ended item questionnaire was: "The following topics regarding ageing in the city should be researched in the future" [translation by the authors, original phrase in German: "Folgende Themen sollten zum Altwerden in der Stadt zukünftig erforscht werden"]. By giving no research question examples or topical suggestions, we aimed to achieve a variety of research themes. In the beginning of October 2021, we sent the study information to each citizen by mail, including one page with the single item questionnaire, and a stamped return envelope. We followed up with a reminder a week later, and another reminder two weeks after the initial letter was sent. The citizens had the possibility to anonymously submit their response by three different means: 1) respond via mail by using the stamped return envelope, 2) call us on a telephone hotline and leave a message on an answering machine, and 3) by using a website with the same open-ended questionnaire item allowing for direct text input in a text box. In case of phone calls, answers were transcribed into text by the second author.

Data analysis was conducted in six steps, of which the first four steps were conducted in a team (first, second, and last author). Those steps included 1) data familiarization through repeated reading and assurance to understand all answers, 2) splitting the answers into units of meaning, 3) constructing at least one code for each unit of meaning, 4) clustering of the codes into categories, 5) using the resulting coding scheme to code the whole text corpus, and 6) descriptive statistic of the coding results. For the data analyses, we used MAXQDA 2020 (VERBI Software, 2021). In many cases, one written response addressed several different topics, in these cases the responses were split into several single segments (462 segments in total, median = 4 segments per participant). Not all these segments were eventually classified into categories. For example, 14 of these segments described subjective conceptions on ageing in general, and 19 segments were complaints or opinions on (communal) political topics, so we decided to exclude them from further coding. The exemplary segments in the results section of this article were translated from German by the second author.

For the workshop phase, we sent a fourth letter to all citizens that were initially chosen to participate in the study and invited them to participate in a local workshop in each of the neighborhoods. Additionally, we invited local government representatives and stakeholders (i.e., NGOs) suggested by local government. The aim was to present the clusters with

Table 1. Key figures for the five chosen neighborhoods in Berlin [11,12].

| District | Neighborhood | Inhabitants | Inhabitants 65 years or older | Age 65+ & migration background | Old Age Poverty[a] |
|---|---|---|---|---|---|
| Spandau | Maulbeerallee | 11,620 | 17.0% | 28.9% | 14.2% |
| Marzahn-Hellersdorf | Böhlener Straße | 5,929 | 8.2% | 4.6% | 8.6% |
| Lichtenberg | Hohenschönhausener Straße | 6,058 | 28.4% | 7.8% | 3.2% |
| Reinickendorf | Treuenbrietzener Straße | 11,610 | 19.0% | 19.4% | 8.9% |
| Treptow-Köpenick | Allende II | 4,368 | 36.0% | 4.9% | 1.9% |

[a] (inhabitants aged 65 or older receiving benefits according to the German Social Code XII).

representative codes and relevant quotes and discuss them. The workshops were planned for November and December 2021. Due to increasing COVID-19 incidence in late 2021 in Berlin, only the workshop in Treptow-Köpenick could be conducted face-to-face. For the other neighborhoods, we offered the possibility to discuss the results via online meeting or through telephone calls to reduce risk of COVID-19 transmission.

Participants were informed about the content of the study, data protection and privacy rights. Participants were asked to answer without revealing any identifying information and were informed that by answering anonymously to the survey, implied consent was given to participation in the study and publication of its results. The study was approved by the ethics committee (Ethikkommission der Charité – Universitätsmedizin Berlin, Reference Number: EA1/254/21).

## Results

A total of 163 participants responded to the survey, of which 102 (10.2%) could be included in the evaluation. We received most of the valid responses (n = 78, 76.5%) via mail using the stamped return envelope. 91 (89.2%) of these answers could be assigned to a planning area: 30 (29.4%) came from Allende II (Treptow-Köpenick), 18 (17.6%) from Hohenschönhauser Straße (Lichtenberg), 15 (14.7) from Maulbeerallee (Spandau) and 14 (13.7%) from Treuenbrietzener Straße (Reinickendorf) and Böhlener Straße (Marzahn-Hellersdorf) respectively.

From the material, we constructed 18 codes which were clustered in six categories: Health, Living Environment, Social Issues, Mobility, Prevention, and Accessibility of Information and Communication. Fig 1 gives an overview of the categories, their respective codes and the number of individual participants whose answers contributed to these categories.

### Health

Regarding health topics, responses from participants focused on health care delivery, but also on the adaptation of health care to the needs of older people and specific diseases. Participants mentioned the geographical distance to health services and the accessibility of health services as issues that should be investigated. The following quote demonstrates a specific research question on the distribution of allied health professions like physiotherapy:"*I am 76 years old and had several surgeries last year, which is why I am very limited in my mobility. I would like to become fit again. However, there are not enough opportunities for physiotherapy to work on becoming fit again. Someone should think about this. How much and which kind of physiotherapy is actually needed per citizen depending on age.*" (P14) Similarly, participants mentioned their desire that research on health services should have a stronger focus on older adults' needs with regard to specific diseases and conditions:"*Problems with pain are not taken seriously by doctors. (. . .) In the field of health care, we should research pain in old age.*" (P23)

### Living environment

Respondents suggested further investigation into different topics in terms of lived environment, for example in the field of urban planning, green spaces and their impact on quality of life ("*I would be interested to know about the influence of soil sealing of urban green spaces (. . .) on the quality of life and health of the older urban population.*" P11). This also included comments on appropriate seating (i.e. benches, chairs), availability of public restrooms, and parks. Regarding their housing situation, participants expressed the importance of age-appropriate apartments, proposing to explore forms of assisted care living, the cause of rising rents, and the geographical distance and potentially difficult logistics of buying groceries and other

**Health**
(64 participants)

**Health Care Delivery Design**

In times of staff shortage

Close to home

Doctor-Patient-Relationship

Specialization on Geriatric Medicine

Scheduling appointments

Allied health professions / therapeutic care

**Understanding Medication and Polypharmacy**

**Health Care Policy**

**Disease-specific topics**

**Living Environment**
(49 participants)

**Urban Environment**

**Housing**

**Supply of Daily Needs**

**RESEARCH TOPICS SUGGESTED BY OLDER ADULTS**

**Social Issues**
(48 participants)

**Social Participation**

Neighborhood and Meeting Places

Leisure Activities

Safety / Security

Old-age poverty

**Interpersonal Interaction**

Cross-generational Activities

Social Engagement

**Loneliness**

**Mobility**
(37 participants)

**Individual Traffic**

By Car

By Bicycle

By Foot

**Public Transport**

**Physical Accessibility**

**Prevention**
(27 participants)

**Physical and Mental Activities**

**Nutrition**

**Addictive Substances**

**Accessibility of Information and Communication**
(19 participants)

**Digital vs. Analogue Means of Communication and Interaction**

**Use of Foreign Languages or Jargon**

**Fig 1. Categories and codes deduced from the suggestions of older adults for research topics on ageing well.**

supply of daily needs. Participants wished for further research on adjustments of the environment to their needs to be conducted.

## Social issues

Loneliness was a topic that – despite being a well-established problem in literature – was still a prominent issue for older adults participating in our study, asking how to address the feeling of isolation in old age. Many participants wondered how social participation in the city can be strengthened while also highlighting the need for feeling safe and wondered how poverty in old age influences the possibility of interacting with others, social contacts and quality of life. Meeting places and leisure activities for older adults seem to be frequented regularly and rated as important social hubs. Some participants expressed that they would like to pursue their professional interests ("*Looking for opportunities for successful older people to contribute their knowledge and experience*", P51) or get involved in cross-generational activities and networking ("*Old people should be brought together with young people so that they're not so lonely*", P97).

## Mobility

Research ideas for mobility included suggestions on individual traffic by car, bicycle or by foot. One participant mentioned the aspect of equality in traffic:"*Research should look into a solution to increase traffic safety, where older adults have equal rights on the road."* (P90) Another respondent proposed the idea of bonus programs for older adults who relinquish their driver's license. Few participants wrote about using the bicycle as means of transport, underlining the topic of safety on the road and the issues of sidewalks sometimes being shared by pedestrians and bikers at the same time. For some participants, the city in general did not seem very accessible. Use and accessibility of local public transport was also named a topic to be researched further:"*I would like to use public transport again (e.g., bus, train). However, there is little attention to older adults with walking aids (e.g., buses start too quickly after entering into)."* (P29)

## Prevention

Participants suggested that more research on prevention of diseases should be conducted. Specific research projects suggested were: age-appropriate physical and mental activities, at what age prevention should begin, how nutrition influences older adults' health, but also proposing more research on early screening programs and dealing with substance abuse. A topic mentioned several times was preventing dementia:"*Develop a screening program for early Alzheimer and dementia detection. If possible, include these in general preventive medical check-ups (like colon and breast cancer etc.) for people in certain age groups.*" (P44)

## Accessibility of information and communication

In this category, participants described their difficulties receiving relevant information in their area (i.e., on local age-centered activities, public transport schedules, for example through the internet or by other means). Participants also reported on problems with reading and understanding foreign languages, or medical jargon on their prescribed medication. They proposed to investigate ideas on how to reach older adults through different means (i.e., via doctors, media, and direct communication), underlining that digital information should not be seen as a mandatory information source for everyone.

# Discussion

When asked to suggest research topics from an older adults' perspective, participants expressed their ideas on a wide range of topics that were clustered into six main categories: health, living environment, social issues, mobility, prevention and accessibility of information and communication. Most suggestions focused on topics of health care delivery and the living environment, but also tackled topics like social interaction and loneliness.

While differing in study design, the topics suggested in our research are in line with results of a qualitative study by Walker et al. [7], in which social networks, close relationships with other people in the neighborhood and proximity of services were defined as important by participants. The findings highlight that older adults suggest research topics from what they experience in their vicinity and what matters to them personally in their daily lives. In another study by Röhr et al., three main themes were defined regarding the design of urban environments to promote brain health: social participation, accessibility and proximity of health care, cultural events and public restrooms, as well as possibilities of local recreation and well-being [9]. All these topics were suggested by older adults in our study, who also mentioned the need for further investigation on disease prevention, specifically dementia prevention. Regarding health care topics, the National Ageing Research Institute of the Victorian Department of

Health in Australia described older adults' needs even more extensively than in the health segment of our study and defined important topics together with older adults, such as healthy and active ageing, independent living, sense of community, as well as care in medical crises and during end of life [10]. Self-acceptance and self-growth were found to be important for successful ageing in a study by Reichstadt et al. [5], but our participants did not specifically mention these themes as further research topics regarding urban ageing.

Research on participatory studies in which participants are actively engaged in designing a research agenda in an open format is limited, which makes it difficult to directly compare our results to similar projects. Also, the open format might have posed a potential intellectual challenge to the participants and may exclude older adults with a lower education from participation. Accordingly, only some of the responses were written in form of a definitive research topic. Responses were often phrased as issues and wishes or gave individual opinions on different topics ranging from national politics to general statements on growing old. We excluded the more general opinions that did not express any area for potential scientific research. Nevertheless, we were able to include many responses into our analysis to define categories, even when they were not specifically phrased as a hypothesis. Due to legal and ethical considerations, the letter sent out to the citizens contained five pages of descriptions on privacy and legal statements, which may have been discouraging especially for older adults who have a lower competency in reading. As the study information was only available in German, some citizens that received our mailings may have not understood their content. We received, however, one single answer in Russian which we had translated into German by a fellow researcher from our Institute. While the open format of the study might have been overwhelming for some, the overall response rate with more than 10% is still satisfactory for a postal survey with older adults who are not used to these kinds of projects. The age threshold that we chose as inclusion criteria (age 65 and older) was based on the earliest possible age of retirement in Germany, but it does not acknowledge possible differences in research ideas from the "young-old" adults (55–75 years) and the "old-old" adults (75+ and above) as described by Neugarten [13].

Unfortunately, the pandemic did not allow to conduct the second, workshop-based stage of the project as planned, so instead of five, only one workshop (in Treptow-Köpenick) could be conducted face-to-face. There, the panel consisted of older adults as well as local stakeholders (district officials, social workers) and allowed for a lively discussion on the presented topics, giving participants the chance to connect with each other. For the other four neighborhoods, we offered phone calls to present the results of the study individually and held video conferences with local stakeholders and interested participants. Online and via phone, the participation was not as strong as the local workshop, but it still gave us the possibility to present the results to stakeholders and interested citizens.

## Conclusion

There is a substantial interest of older adults in urban environments on research regarding their living situation, especially focusing on health care delivery, living environment and psychosocial aspects, such as loneliness, interpersonal interaction, and concerns about living conditions. The focus on older adults' daily lives should be more elaborated in the identified categories to promote aging well in cities. The suggested research topics can serve as a base for researchers on which to select themes that need further investigation, and involve different age groups of older adults to better reflect and integrate their perspectives.

## Acknowledgments

We thank all participants and stakeholders for their contributions to this study. We also appreciate the assistance of our colleague Konrad Laker in language editing for the revised manuscript.

## Author Contributions

**Conceptualization:** Philip Oeser, Wolfram J. Herrmann.

**Data curation:** Nora Bruckmann.

**Formal analysis:** Nora Bruckmann.

**Funding acquisition:** Wolfram J. Herrmann.

**Investigation:** Philip Oeser, Nora Bruckmann.

**Project administration:** Philip Oeser, Nora Bruckmann, Wolfram J. Herrmann.

**Supervision:** Paul Gellert, Wolfram J. Herrmann.

**Writing – original draft:** Philip Oeser, Nora Bruckmann.

**Writing – review & editing:** Philip Oeser, Paul Gellert, Wolfram J. Herrmann.

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
