## [Decision Letter · Decision Letter 0]

29 Aug 2023

PONE-D-23-22761Older adults’ suggestions of research topics on ageing well in urban environments – a participatory studyPLOS ONE

Dear Dr. Oeser

Thank you for submitting your manuscript to PLOS ONE. After careful consideration, we feel that it has merit but does not fully meet PLOS ONE’s publication criteria as it currently stands. Therefore, we invite you to submit a revised version of the manuscript that addresses the points raised during the review process.

We look forward to receiving your revised manuscript.

Kind regards,

Grant Rich, Ph.D.

Academic Editor

PLOS ONE

Journal Requirements:

2. Please provide additional details regarding participant consent. In the ethics statement in the Methods and online submission information, please ensure that you have specified what type you obtained (for instance, written or verbal, and if verbal, how it was documented and witnessed). If your study included minors, state whether you obtained consent from parents or guardians. If the need for consent was waived by the ethics committee, please include this information."

3. Our internal editors have looked over your manuscript and determined that it is within the scope of our Aging in Human Health and Disease Call for Papers. This call for papers aims to highlight the excellent work being done by researchers across the world on the subject of aging. Additional information can be found on our announcement page: https://collections.plos.org/call-for-papers/aging-in-human-health-and-disease/. If accepted, your submission will be included within the collection. Please note that being considered for the Collection does not require an additional peer review beyond the journal’s standard process and will not delay the publication of your manuscript if it is accepted by PLOS ONE. If you have any questions or concerns about this process, please contact the journal at plosone@plos.org.

Additional Editor Comments (if provided):

Your topic is excellent and significant and the methods appropriate. Please submit a minor revision- I suggest you consult an English speaking helper to improve a few passages in the writing that may be improved for clarity and fluency. Please address the comments of both reviewers below- for reviewer two you may wish to consult or cite classic lifespan development work by Bea Neugarten at the University of Chicago on "young old" vs "old old" to distinguish say, persons aged 55 to 65 vs those 85 to 95, and perhaps her work on "social clock" and on time vs "off time" development as well as work by her modern colleague at U Chicago Rick (Richard) Shweder's work on Middle Age and aging from cultural construction/anthropological view.

---

REVIEWER 1

Accept

While not a quantitative study, the numerical data makes sense and aligns with qualitative practices.

Please update quotation marks to standard US English formatting. Check the use of "sealing" and "seating" on lines 169 and 171. Are these supposed to be the same word?

REVIEWER 2 Major revision

The idea of looking at "older adults health needs" is very valuable, but it seems that the inclusion of all participants in the age brackets (65 +) can be confusing. Authors should clarify if they are combining all "older adults"? If so, this can be problematic because the needs of this population and as such research needs can vary according to the specific chronological ages. That is a 65-year-old person's needs can be quite different than an eighty-year-old, although there could be important similarities. Clarification of the cohort groups and what are the common characteristics they share is required. Additionally, and because of these different characteristics of the cohort groups, need to be defined more clearly. It is recommended that the authors strengthen the categories with a clearer definition of the different ages and specific needs.

Reviewers' comments:

Reviewer's Responses to Questions

**Comments to the Author**

1. Is the manuscript technically sound, and do the data support the conclusions?

Reviewer #1: Yes

Reviewer #2: Partly

2. Has the statistical analysis been performed appropriately and rigorously? 

Reviewer #1: Yes

Reviewer #2: I Don't Know

3. Have the authors made all data underlying the findings in their manuscript fully available?

Reviewer #1: Yes

Reviewer #2: Yes

4. Is the manuscript presented in an intelligible fashion and written in standard English?

Reviewer #1: Yes

Reviewer #2: Yes

5. Review Comments to the Author

Reviewer #1: While not a quantitative study, the numerical data makes sense and aligns with qualitative practices.

Please update quotation marks to standard US English formatting. Check the use of "sealing" and "seating" on lines 169 and 171. Are these supposed to be the same word?

Reviewer #2: The idea of looking at "older adults health needs" is very valuable, but it seems that the inclusion of all participants in the age brackets (65 +) can be confusing. Authors should clarify if they are combining all "older adults"? If so, this can be problematic because the needs of this population and as such research needs can vary according to the specific chronological ages. That is a 65-year-old person's needs can be quite different than an eighty-year-old, although there could be important similarities. Clarification of the cohort groups and what are the common characteristics they share is required. Additionally, and because of these different characteristics of the cohort groups, need to be defined more clearly. It is recommended that the authors strengthen the categories with a clearer definition of the different ages and specific needs.

6. PLOS authors have the option to publish the peer review history of their article (what does this mean?). If published, this will include your full peer review and any attached files.

Reviewer #1: **Yes: **Stephanie Elizabeth Beckman

Reviewer #2: No

Grant J. Rich, PhD 

 Candidate for President-Elect for the American Psychological Association Twitter/X: @GrantJRich4APA 

President-Elect Society for Peace, Conflict, and Violence (APA)
 President-Elect Society for Media Psychology and Technology (APA)

Fellow, Association for Psychological Science (APS)
 Fellow, American Psychological Association (APA) (D1, D2, D46, D48, D52)

Senior Contributing Faculty, Walden University
 Editorial Board Member: PLOS ONE, APA's Peace & Conflict,
APA's Traumatology

Book Series Co-Editor w/ Anthony Marsella (U. Hawai'i), Springer. International and Cultural Psychology (ICUP)       

 https://www.springer.com/series/6089 Select Recent Books

(Rich, Gielen, & Takooshian,
2017).
Internationalizing the Teaching of Psychology.

IAP.

(Rich & Sirikantraporn, 2018).
Human Strengths and Resilience: Cross Cultural and International Perspectives. Rowman & Littlefield.. 

(Rich, Jaafar, & Barron, 2020).
Psychology in Southeast Asia. Routledge.

(Rich & Ramkumar, 2022).
Psychology in Oceania and the Caribbean. Springer. 

(Rich, Kuriansky, Gielen, & Kaplan, 2023).
Psychosocial Experiences and Adjustment of Migrants: Coming to the USA. 
Elsevier. 

(Rich, Kumar, & Farley, in contract).
Handbook of Media Psychology and Technology-The Science and the Practice. Springer.

---

## [Author Response · Author response to Decision Letter 0]

7 Sep 2023

Response to Reviewers

Dear editor and reviewers, 

Thank you very much for considering our manuscript and taking the time to give valuable comments. We implemented all your revisions and updated the manuscript accordingly. Furthermore, we corrected a small error in the number of participants that we found during revision. We now think that the manuscript has greatly improved and hope that it is acceptable for publication. Below, we reply to your comments point by point. 

EDITOR

1. Please ensure that your manuscript meets PLOS ONE's style requirements, including those for file naming. The PLOS ONE style templates can be found at ...

We checked the manuscript to meet PLOS ONE’s style requirements according to your templates and updated the file names prior to uploading the revision.

2. Please provide additional details regarding participant consent. In the ethics statement in the Methods and online submission information, please ensure that you have specified what type you obtained (for instance, written or verbal, and if verbal, how it was documented and witnessed). If your study included minors, state whether you obtained consent from parents or guardians. If the need for consent was waived by the ethics committee, please include this information."

Participants were asked to answer the questionnaire anonymously and were informed that by answering, implied consent was given to participate and for publication of the results. We updated the methods section and the online submission information accordingly.

3. Our internal editors have looked over your manuscript and determined that it is within the scope of our Aging in Human Health and Disease Call for Papers. […]

We are grateful that our manuscript is being considered for the Aging in Human Health and Disease Call for Papers and that we can contribute to the collection.

We made three changes to the reference list: we updated the format of reference no. 1 and no. 2 and added URLS to the respective databases. Also, reference no. 13 (Neugarten BL. Age Groups in American Society and the Rise of the Young-Old) was added. 

While revising your submission, please upload your figure files to the Preflight Analysis and Conversion Engine (PACE) digital diagnostic tool, https://pacev2.apexcovantage.com/. 

The figure we use in the manuscript is now processed by PACE and will be re-uploaded in the new format.

Additional Editor Comments (if provided):

Your topic is excellent and significant and the methods appropriate. Please submit a minor revision - I suggest you consult an English speaking helper to improve a few passages in the writing that may be improved for clarity and fluency. 

We have revised the manuscript together with an English speaking colleague to improve text flow and clarity.

---

REVIEWER 1

Please update quotation marks to standard US English formatting. 

Thank you for your revisions on the manuscript. As part of the language editing for this revision, we corrected the passages in which we are using quotation marks to fit the US English formatting.

Check the use of "sealing" and "seating" on lines 169 and 171. Are these supposed to be the same word?

The first quote mentioning “sealing” refers to the sealing of soil in urban green spaces, while the second quote “seating” refers to seating possibilities. We changed the wording in the manuscript to make the difference clearer (lines 179-181 in the manuscript with mark-ups).

---

REVIEWER 2

The idea of looking at "older adults health needs" is very valuable, but it seems that the inclusion of all participants in the age brackets (65 +) can be confusing. Authors should clarify if they are combining all "older adults"? If so, this can be problematic because the needs of this population and as such research needs can vary according to the specific chronological ages. That is a 65-year-old person's needs can be quite different than an eighty-year-old, although there could be important similarities. Clarification of the cohort groups and what are the common characteristics they share is required. Additionally, and because of these different characteristics of the cohort groups, need to be defined more clearly. It is recommended that the authors strengthen the categories with a clearer definition of the different ages and specific needs.

Thank you very much for this important comment. Due to data security reasons, we decided not to ask for additional demographic data of the participants, such as age and socioeconomic status to keep the data anonymous. We acknowledge that the needs and research ideas of different age groups can be quite different. Thus, we added a paragraph in the limitations sections of the manuscript describing this inherent limitation of our study.

Kind regards, 

Philip Oeser on behalf of all the authors

---

## [Editor Report · Decision Letter 1]

18 Sep 2023

Older adults’ suggestions of research topics on ageing well in urban environments – a participatory study

PONE-D-23-22761R1

Dear Drs Oeser, Bruckmann, Gellert, and Herrmann

We’re pleased to inform you that your manuscript has been judged scientifically suitable for publication and will be formally accepted for publication once it meets all outstanding technical requirements.

Kind regards,

Grant Rich, Ph.D.

Academic Editor

PLOS ONE

Additional Editor Comments (optional):

Dear Authors, you have now made the required revisions, and I am happy to accept this valuable article for publication, Dr Rich

Reviewers' comments:

Grant J. Rich, PhD 

Candidate for President-Elect for the American Psychological Association

Twitter/X: @GrantJRich4APA

President-Elect Society for Peace, Conflict, and Violence (APA)

President-Elect Society for Media Psychology and Technology (APA)

Fellow, Association for Psychological Science (APS)

Fellow, American Psychological Association (APA) (D1, D2, D46, D48, D52)

Senior Contributing Faculty, Walden University

Editorial Board Member: PLOS ONE, APA's Peace & Conflict,
APA's Traumatology

Book Series Co-Editor w/ Anthony Marsella (U. Hawai'i), Springer. International and Cultural Psychology (ICUP)       

 https://www.springer.com/series/6089

Select Recent Books

(Rich, Gielen, & Takooshian,
2017).
Internationalizing the Teaching of Psychology.

IAP.

(Rich & Sirikantraporn, 2018).
Human Strengths and Resilience: Cross Cultural and International Perspectives. Rowman & Littlefield.

(Rich, Jaafar, & Barron, 2020).
Psychology in Southeast Asia. Routledge.

(Rich & Ramkumar, 2022).
Psychology in Oceania and the Caribbean  (Foreword by past APA President Frank Worrell).Springer. 

.(Foreword by past APA President Tony Puente).(Rich, Kuriansky, Gielen, & Kaplan, 2023)
Psychosocial Experiences and Adjustment of Migrants: Coming to the USA

Elsevier.

(Rich, Kumar, & Farley, in contract).
Handbook of Media Psychology and Technology-The Science and the Practice. Springer.

---

## [Editor Report · Acceptance letter]

25 Sep 2023

PONE-D-23-22761R1 

Older adults’ suggestions of research topics on ageing well in urban environments – a participatory study 

Dear Dr. Oeser:

I'm pleased to inform you that your manuscript has been deemed suitable for publication in PLOS ONE. Congratulations! Your manuscript is now with our production department. 

Kind regards, 

on behalf of

Dr. Grant Rich 

Academic Editor

PLOS ONE